# Observing geometry effects on a GNSS based water vapor tomography solved by Least Squares and by Compressive Sensing

Marion Heublein[1], Patrick Erik Bradley[1], and Stefan Hinz[1]

[1]Karlsruhe Institute of Technology, Institute of Photogrammetry and Remote Sensing, 76128 Karlsruhe, Germany

**Correspondence:** Marion Heublein (marion.heublein@kit.edu)

**Abstract.** In this work, the effect of the observing geometry on the tomographic reconstruction quality of both a regularized Least Squares (LSQ) and a Compressive Sensing (CS) approach for water vapor tomography is compared based on synthetic Global Navigation Satellite System (GNSS) Slant Wet Delay (SWD) estimates. In this context, the term observing geometry mainly refers to the number of GNSS sites situated within a specific study area subdivided into a certain number of volumetric pixels (voxels) and to the number of signal directions available at each GNSS site. The novelties of this research are 1) the comparison of the observing geometry's effects on the tomographic reconstruction accuracy when using LSQ resp. CS for the solution of the tomographic system and 2) the investigation of the effect of the signal directions' variability on the tomographic reconstruction. The tomographic reconstruction is performed based on synthetic SWD data sets generated, for many samples of various observing geometry settings, based on wet refractivity information from the Weather Research and Forecasting (WRF) model. The validation of the achieved results focuses on a comparison of the refractivity estimates with the input WRF refractivities. The results show that the recommendation of Champollion et al. (2004) to discretize the analyzed study area into voxels with horizontal sizes comparable to the mean GNSS inter site distance represents a good rule of thumb for both LSQ and CS based tomography solutions. In addition, this research shows that CS needs a variety of at least 15 signal directions per site in order to estimate the refractivity field more accurately and more precisely than LSQ. Therefore, the use of CS is particularly recommended for water vapor tomography applications for which a high number of multi-GNSS SWD estimates are available.

## 1 Introduction

An accurate determination of the three dimensional (3D) atmospheric water vapor distribution is essential for weather forecasting and climate research. In addition, atmospheric water vapor delays the microwave signal propagation within the atmosphere and thus represents an error source in e.g. Global Navigation Satellite Systems (GNSS) and Interferometric Synthetic Aperture Radar (SAR) (InSAR) observations. Therefore, a precise knowledge of the water vapor field e.g. is required for accurate deformation monitoring using InSAR. However, the atmospheric water vapor distribution is difficult to model because it is highly variable in time and space. Several approaches exist for reconstructing the 3D tomographic water vapor reconstruction using one dimensional (1D) GNSS SWDs, see Section 2.

One of the main limiting factors in water vapor tomographies consists in the point-wise GNSS observing geometry, which

causes an ill-conditioned inverse tomographic model that needs to be regularized. Yet, even after regularization, the observing geometry composed e.g. of the number and the geographic distribution of the GNSS sites, the SWD signal directions, and the voxel discretization still effects the quality of the tomographic solution. This work therefore meets the challenge of comparing the observing geometry's effect on a GNSS based water vapor tomography solved by means of Least Squares (LSQ) resp. by means of Compressive Sensing (CS). By investigating the observing geometry's effect on the LSQ and CS solution strategies, the differences between the LSQ solution and a CS solution approach benefiting of the signal's sparsity in an appropriate transform domain for regularization are better understood and recommendations can be given for future water vapor tomography campaigns and the processing of their measurements. Based on synthetic data sets deduced from the Weather Research and Forecasting Model (WRF) described in Skamarock et al. (2005), the presented work answers the research questions 1) in how far the rule of thumb of Champollion et al. (2004), derived for LSQ and recommending a voxel size corresponding to the mean GNSS inter site distance, can be transferred from a LSQ solution to a CS solution. In addition, this research investigates 2) in which settings CS is able to more accurately and more precisely reconstruct the tomographic water vapor field than LSQ and 3) to which extent multi-GNSS SWD observations improve the tomographic solution obtained by means of LSQ resp. CS, when compared to solutions obtained from SWDs originating from the Global Positioning System (GPS) only.

## 2  Related work

Current water vapor tomographies can be distinguished e.g. based on the methodology and the data sets applied for solving the tomographic model. The tomographic model is commonly established based on the directions along which space-geodetic SWD estimates are acquired and based on a discretization of the investigated atmospheric volume into volumetric pixels (voxels) e.g. of constant refractivity. The existing tomography solution approaches applied to such a discretized atmosphere are subdivided into iterative and non-iterative techniques. Bender et al. (2011a) propose different iterative Algebraic Reconstruction Techniques (ART), while Hirahara (2000), Flores et al. (2000), Champollion et al. (2004), Troller (2004), Song et al. (2006), Notarpietro et al. (2008), and Rohm (2013) apply different non-iterative methods for solving the tomographic system using a LSQ adjustment. Thanks to its good capability to estimate dynamically changing parameters, Flores et al. (2000), Gradinarsky and Jarlemark (2004), and Rohm et al. (2014) choose a Kalman filter approach. Hirahara (2000) proposes a damped least squares solution known from seismic tomography to solve the tomographic problem. Xia et al. (2013) combine iterative and non-iterative techniques. Instead of using voxels in water vapor tomographies with a small number of GNSS sites, Ding et al. (2018) discretize the tomographic field based on the perimeter of the tomographic boundary on the plane and based on meshing techniques. They then determine tomographic fields by means fitting the real distribution of GNSS signals on different tomographic planes at different tomographic epochs.

In addition to slant wet delay estimates from GNSS, Hurter and Maier (2013) introduce wet refractivity profiles from radio occultation and radiosonde observations into a combined least squares collocation. Rather than using slant wet delay estimates as input observations, Nilsson and Elgered (2007) apply a solution that relies directly on GPS phase observations.

Independently of the reconstruction strategy, due to the point-wise GNSS observing geometry, the tomographic system of

equations is usually ill-posed and needs to be regularized e.g. i) by constraining the tomographic system by means of pseudo observations, ii) by introducing additional observations from models, from simulations, or from other sensors, or iii) by decreasing the amount of voxels crossed by no rays at all.

Both Flores et al. (2000) and Gradinarsky and Jarlemark (2004) regularize the solution by means of adding horizontal and vertical smoothing constraints to the tomographic system and by means of introducing a boundary constraint assuming the refractivity to approach zero above a certain height. Alternatively, as proposed in Elosegui et al. (1998), the refractivity field can be assumed to decrease exponentially with increasing height. Yet, while regularizing the solution significantly, both geometric constraints and the exponential decay usually are not able to accurately model the real atmospheric state. As an alternative to using horizontal and vertical constraints relying on physical approximations to the atmospheric behavior, the work in Heublein et al. (2018) and Heublein (2019) exploit the signal's sparsity in a particular, predefined transform domain as prior knowledge for regularizing the tomographic system in order to then reconstruct the signal using an $L_1$-norm minimization. Similarly, CS and sparse reconstruction are applied here for the tomographic reconstruction of the 3D water vapor field and the CS solution to water vapor tomography is compared to a solution obtained using a classical LSQ approach. Initially presented by Candès et al. (2006), Donoho (2006), Baraniuk (2007), and Candès and Wakin (2008) for the image or signal recovery from a number of samples below the desired resolution or the Nyquist rate, CS has been, since then, applied to many remote sensing problems in which sparse signals occur. E.g. Potter et al. (2010) and Alonso et al. (2010) describe the use of CS for SAR imaging, Pruente (2010) applies CS for ground moving target identification, Zhu and Bamler (2010), Budillon et al. (2011), Aguilera et al. (2013), and Zhu and Bamler (2014) apply CS to SAR tomography, and Li and Yang (2011), Zhu and Bamler (2013), Grohnfeldt et al. (2013), Jiang et al. (2014), and Zhu et al. (2016) use CS for pan-sharpening and hyperspectral image enhancement. When compared to classical LSQ adjustments usually applying $L_2$-norm regularizations, Compressive Sensing and sparse reconstruction based on a small number of measurements led to promising results.

However, Compressive Sensing only yields encouraging results if the input data acquisition – corresponding, in water vapor tomography, to the determination of SWD estimates – fulfills certain prerequisites. For general applications of CS, Rauhut (2010) e.g. states that randomness in the acquisition step helps to utilizing the minimum number of measurements. When reconstructing images based on frequency data, e.g. Candès et al. (2006) alternatively recommend to randomly measure frequency coefficients such that sparse objects are sensed by taking as few measurements as possible. For CS-based water vapor tomography approaches, no explicit requirements for the SWD acquisition or for designing advantageous observing geometry settings have been established so far.

For LSQ, Champollion et al. (2004) state that the optimal horizontal size of a voxel should correspond to the mean inter-site distance between the used GNSS sites. Given a certain cutoff elevation angle, the height layers' thicknesses in their approach should be defined such that signals received at a GNSS site situated within a voxel's center are able to cross neighboring voxels. Due to the small wet refractivity values in the upper layers and in order to make the tomographic solutions less sensitive to errors in the input data, Rohm (2012) recommends to increase the height layer thicknesses with increasing altitude. Bender and Raabe (2007) resp. Bender et al. (2009) estimate the spatial distribution of the geometric intersection points between different ray paths in order to compute the information density contained in a given set of GPS signals. They then use this information

as a precondition to an optimal tomographic reconstruction resp. in order to identify regions that are well covered by GPS slant paths. Although Bender et al. (2011b) and Zhao et al. (2019) state that changing the observing geometry by combining multi-GNSS observations instead of GPS only observations does not substantially improve the reconstruction quality, Rohm (2012) realizes that the uncertainty of the tomographic solution is largely influenced by the mathematical properties of the design matrix, depending itself on the observing geometry. With the aim of giving advice for the installation of new permanent sites and for the solution of future water vapor tomographies, this work therefore investigates the observing geometry's effect on the quality of both a LSQ and a CS solution to the tomographic system.

## 3   Methodology

In order to analyze the observing geometry's effect on the quality of the LSQ and CS solution to water vapor tomography, different observing geometry settings are defined. Based on synthetic SWD estimates derived from WRF, 3D water vapor distributions are reconstructed for each of the defined observing geometry settings using both LSQ and CS. The quality of the LSQ and CS solutions to water vapor tomography is then compared w.r.t. the respective observing geometry settings.

### 3.1   Tomographic model

For tomography using GNSS SWDs, Flores et al. (2000) introduce the functional model

$$\mathrm{SWD}_{i,\,\mathrm{cont}} = 10^{-6} \cdot \int\limits_{\mathrm{sp}_i} N_{\mathrm{wet}} \, \mathrm{d}l, \tag{1}$$

where $\mathrm{SWD}_{i,\,\mathrm{cont}}$ stands for the integrated slant wet delay observations between a certain satellite and a certain GNSS site and $\mathrm{d}l$ is a differential along the slant ray path. As in Heublein et al. (2018) and Heublein (2019), the variable $\mathrm{sp}_i$ is the $i$th slant ray path, i.e. the slant ray path of the radiowave signal between a certain satellite and a certain receiver. The variable $N_{\mathrm{wet}}$ contains the wet refractivity along this path. The index $i$ attains the values

$$i \in \mathbb{N} \text{ with } 1 \leq i \leq N, \tag{2}$$

where $N$ corresponds to the number of observations available between any receiver and any satellite. When tomographically reconstructing the wet refractivity, however, the continuous functional model from Equation 1 is usually replaced by a discrete functional model

$$\mathrm{SWD}_{i,\,\mathrm{disc}} = 10^{-6} \cdot \sum_{j=1}^{L} N_{\mathrm{wet}\,j} \cdot d_{ij}. \tag{3}$$

That is, the 3D water vapor distribution is discretized into $L = \mathcal{P} \times \mathcal{Q} \times \mathcal{K}$ voxels in longitude, latitude, and height, assuming a constant refractivity value for each voxel. As in Heublein et al. (2018) and Heublein (2019), in this work, a uniform voxel discretization is selected in the horizontal directions, while the voxel sizes in the vertical direction increase with increasing height. Horizontally, the voxels are limited by constant longitudes and latitudes. In the vertical direction, the voxels are separated by layers of constant ellipsoidal height.

As in Heublein et al. (2018) and Heublein (2019), summarizing all observations $\text{SWD}_{i,\text{disc}}$ in an observation vector $\boldsymbol{y}_\text{data} \in \mathbb{R}^{N \times 1}$, all unknown refractivities $N_{\text{wet}\,j}$ for

$$j \in \mathbb{N} \text{ with } 1 \leq j \leq L \tag{4}$$

in a parameter vector $\boldsymbol{x} \in \mathbb{R}^{L \times 1}$, and all distances $d_{ij}$ in a design matrix $\boldsymbol{\Phi}_\text{data} \in \mathbb{R}^{N \times L}$, the discrete tomographic system from Equation 3 can be rewritten as

$$\boldsymbol{y}_\text{data} = \boldsymbol{\Phi}_\text{data} \cdot \boldsymbol{x} \tag{5}$$

with

$$\Phi_\text{data}(i,j) = \begin{cases} d_{ij} & \text{if signal } i \text{ crosses voxel } j \\ 0 & \text{otherwise.} \end{cases} \tag{6}$$

As each signal only passes a small subsection of the study area, most entries of the matrix $\boldsymbol{\Phi}_\text{data}$ are zero and only a few matrix elements are non-zero (e.g. only about $5\,\%$ of the entries of $\boldsymbol{\Phi}_\text{data}$ are non-zero in the case of an about $95 \times 99\,\text{km}^2$ large study area subdivided into $5 \times 5 \times 5$ voxels, disposing of seven GNSS sites and ten signals per site). For voxels that are not crossed by any signals, $\boldsymbol{\Phi}_\text{data}$ has a zero column. Therefore, the tomographic model and the mathematical properties of the design matrix largely depend on the observing geometry settings described in Section 3.3.

## 3.2 Solution of the inverse tomographic model using LSQ resp. CS

The LSQ solution to Equation 5 is derived by solving the minimization problem

$$\hat{\boldsymbol{x}} = \underset{\boldsymbol{x}}{\arg\min} \left\{ \underbrace{\|\boldsymbol{y}_\text{data} - \boldsymbol{\Phi}_\text{data} \cdot \boldsymbol{x}\|_2^2}_{\text{data fidelity term}} + \underbrace{\sum_{t=1}^{3} \Gamma_{\text{constraints}_t}^2 \cdot \|\boldsymbol{y}_{\text{constraints}_t} - \boldsymbol{\Phi}_{\text{constraints}_t} \cdot \boldsymbol{x}\|_2^2}_{\text{regularization constraints and prior knowledge}} \right\}, \tag{7}$$

regularized by means of $t = 3$ regularization terms, namely by horizontal and vertical smoothing constraints as well as by prior knowledge from surface meteorology. Namely, the wet refractivity at the surface is computed from in-situ observations of pressure, temperature, and dew point temperature at a single weather site within study area. As described in Heublein et al. (2018), the horizontal smoothing constraints assuming the refractivity of a voxel $(a,b,k)$ to equal the weighted mean refractivity of the surrounding voxels $(p,q,k)$ with voxel indices $p \neq a$ and $q \neq b$ within the same height layer $k$ are defined by:

$$N_{\text{wet}_{a,b,k}} = \sum_{p,q} w_{p-a,q-b} \cdot N_{\text{wet}_{p,q,k}} \tag{8}$$

The weights can e.g. be derived using inverse distance weighting

$$
\quad w_{p-a,q-b} = \begin{cases} \dfrac{\dfrac{1}{d_{p-a,q-b}}}{\sum\limits_{p,q} \dfrac{1}{d_{p-a,q-b}}} & \text{if } (a,b) \neq (p,q) \\[2ex] -1 & \text{if } (a,b) = (p,q), \end{cases} \tag{9}
$$

with distances $d_{p-a,q-b}$ between the center of voxel $(p,q)$ and the center of voxel $(a,b)$ of the considered $k$th height layer.
Moreover, Davis et al. (1993) state that an average refractivity profile can be approximated assuming the refractivity to expo-
nentially decrease with height:

$$
\quad N_{\text{wet}}(h_k) = N_{\text{wet}}(h_0) \cdot \exp\left(-\frac{h_k - h_0}{H_{\text{scale}}}\right) \tag{10}
$$

The variable $h_k$ is the height of the $k$th layer, $h_0$ stands for some reference height at which the refractivity equals $N_{\text{wet}}(h_0)$, and
$H_{\text{scale}}$ represents the scale height of the local troposphere. As $H_{\text{scale}}$ is essential for defining an exponential decay with height,
its value is determined within the solution of the tomographic system from a set of realistic values for $H_{\text{scale}}$ between 1000 m
and 2000 m. Both the weights for the horizontal and vertical smoothing constraints and for the prior knowledge from surface
meteorology are determined w.r.t. the data fidelity term using the place holder trade-off parameter $\Gamma^2_{\text{constraint}}$ in Equation 7.
The selection of the trade-off parameters from a certain number of logarithmically scaled possible trade-off parameters and the
selection of $H_{\text{scale}}$ are described in Heublein et al. (2018) and Heublein (2019).
When aiming at a tomographic reconstruction of atmospheric water vapor by means of Compressive Sensing, the parameters
$x$ are sparsely represented in some transform domain

$$
\quad \boldsymbol{x} = \boldsymbol{\Psi} \cdot \boldsymbol{s} \tag{11}
$$

as sparse parameters $\boldsymbol{s}$. Estimates $\hat{s}$ for these sparse parameters are obtained by

$$
\quad \hat{\boldsymbol{s}} = \underset{\boldsymbol{s}}{\arg\min}\left\{ \underbrace{\|\boldsymbol{y} - \boldsymbol{\Phi} \cdot \boldsymbol{\Psi} \cdot \boldsymbol{s}\|_2^2}_{\text{data fidelity term}} + \underbrace{\Gamma^2_{\text{CS}} \cdot \|\boldsymbol{s}\|_1}_{L_1-\text{norm regularization}} + \underbrace{\Gamma^2_{\text{constraints}} \cdot \|\boldsymbol{y}_{\text{constraints}} - \boldsymbol{\Phi}_{\text{constraints}} \cdot \boldsymbol{\Psi} \cdot \boldsymbol{s}\|_2^2}_{\text{prior knowledge from surface meteorology}} \right\} \tag{12}
$$

as described in Heublein et al. (2018) and Heublein (2019). Instead of adding horizontal and vertical constraints to the data
fidelity term as in Equation 7, an $L_1$-norm regularization term is introduced in the CS solution to promote sparse solutions for
$\boldsymbol{s}$, as described in Heublein et al. (2018) and Heublein (2019). The $L_1$-norm minimization of the sparse parameters reduces the
solution space. The wet refractivity estimates $\hat{\boldsymbol{x}}$ are then reconstructed using

$$
\quad \hat{\boldsymbol{x}} = \boldsymbol{\Psi} \cdot \hat{\boldsymbol{s}} \tag{13}
$$

with a dictionary $\boldsymbol{\Psi} \in \mathbb{R}^{L \times M}$. As in Heublein et al. (2018) and Heublein (2019), the dimension $M$ of the parameters $\boldsymbol{s} \in \mathbb{R}^{M \times 1}$
in the transform domain varies with the number of base functions resp. atoms in $\boldsymbol{\Psi}$. Similarly to Heublein et al. (2018) and

Heublein (2019), we assert that a sparse representation of the refractivity distribution can be obtained by means of e.g. a dictionary composed of Kronecker products of inverse Discrete Cosine Transform (iDCT) letters in longitude and latitude directions and of Euler letters and Dirac letters in the height direction. When thinking of languages, an atom would stand for a word within a dictionary. Comparably to a word composed of different letters within a language dictionary, each atom results from the Kronecker product of smaller items, namely letters, within the dictionary for sparse representation. I.e., for each column, i.e. line, of the three-dimensional (3D) wet refractivity signal, a Kronecker product of the one-dimensional (1D) letters along the longitude, latitude, and height direction are computed. In the longitude and latitude directions, iDCT letters shall represent horizontal refractivity variations. The Euler letters model an exponential refractivity decay with height, and the Dirac letters describe deviations from a decay described by a linear combination of Euler letters.

## 3.3 Observing geometry settings

In Section 4, tomographic solutions obtained based on a high number of different observing geometry settings are compared. The observing geometry settings result from i) a fixed voxel discretization, ii) seven to 32 sites, iii) five to 20 signal directions per site, and iv) 48 signal direction samples per number of sites and signals. Champollion et al. (2004) recommends i) horizontal voxel sizes for a LSQ solution to water vapor tomography greater than or equal to the mean inter-site distance between the available GNSS sites, i.e. voxel sizes greater than or equal to about $37 \times 37\,\mathrm{km}^2$ resp. to about $17 \times 17\,\mathrm{km}^2$ in the case of seven resp. 32 uniformly distributed GNSS sites within the investigated study area of about $95 \times 99\,\mathrm{km}^2$ size. In this work, the study area is discretized into $5 \times 5 \times 5$ voxels of horizontal sizes of about $19 \times 20\,\mathrm{km}^2$. In the vertical direction, five height layers are distinguished. With increasing height, the height layer thicknesses increase from $1300\,\mathrm{m}$ up to $2900\,\mathrm{m}$. The lowest layer's thickness is set to $1300\,\mathrm{m}$ in order to ensure at least for signals with very low elevation angles that a signal arriving at the center of a voxel is able to pass the horizontally neighboring voxel within the same height layer. This is only possible if the minimum thickness $\Delta h_{\mathrm{min}}$ of the height layers is related to the horizontal voxel size $\Delta\mathrm{hz} = 20\,\mathrm{km}$ and to the cutoff elevation angle $\epsilon_{\mathrm{cut}} = 7°$ by means of

$$\Delta h_{\mathrm{min}} = \frac{1}{2} \cdot \Delta\mathrm{hz} \cdot \tan\epsilon_{\mathrm{cut}}. \tag{14}$$

The ii) minimum number of seven sites originates from the real GNSS Upper Rhine Graben (URG) network site distribution within the analyzed study area. The maximum number of sites is chosen such that the rule of thumb of Champollion et al. (2004), introduced in Section 2, is clearly fulfilled. The horizontal position of the synthetic GNSS sites corresponds, for seven sites, to the position of real GNSS sites within the analyzed study area. The horizontal position of the additionally defined synthetic GNSS sites is chosen such that they are uniformly distributed within the study area. The vertical position of the synthetic GNSS sites corresponds to the height of the WRF digital elevation model at the horizontal position of the sites. The iii) number of signal directions per site is motivated by the GPS resp. by a multi-GNSS orbit geometry. According to Feairheller and Clark (2006), the Global'naya Navigatsionnaya Sputnikova Sistema (GLONASS) constellation disposes of 21 active plus three spare satellites on three orbital planes inclined by $64.8°$ w.r.t. the equator. In contrast, Hofmann-Wellenhof et al. (2008) describe the GPS orbit constellation to consist of 21 satellites plus three spares on six orbital planes inclined by

55° w.r.t. the equator, and the Galileo orbit constellation to contain 27 operational plus three spare satellites on three planes inclined by 56° w.r.t. the equator. Therefore, five signal directions resp. five visible satellites correspond to a pessimistic GPS setting e.g. with site specific shadowing. Eight signal directions per site may be considered as a typical GPS setting at a GNSS permanent site, at which about 30 % of the total number of GPS satellites is visible at a time. Similarly, a total number of 20 signal directions per site corresponds to a visibility of about 30 % of the total number of satellites at a time within a multi-GNSS constellation composed of GPS, GLONASS, and Galileo. For each of the mentioned numbers of sites and numbers of visible satellites, 48 signal direction samples are defined. Given the GPS repeat cycle of about one day, the number of 48 signal direction samples is chosen in order to emulate about half-hourly orbit samples. Using the orbit characteristics mentioned above, synthetic satellite positions are approximated by means of circular orbits. The ii) minimum number of seven sites originates from the real GNSS URG network site distribution within the analyzed study area. The maximum number of sites is chosen such that the rule of thumb of Champollion et al. (2004), introduced in Section 2, is clearly fulfilled. The horizontal position of the synthetic GNSS sites corresponds, for seven sites, to the position of real GNSS sites within the analyzed study area. The horizontal position of the additionally defined synthetic GNSS sites is chosen such that they are uniformly distributed within the study area. The vertical position of the synthetic GNSS sites corresponds to the height of the WRF digital elevation model at the horizontal position of the sites. The iii) number of signal directions per site is motivated by the GPS resp. by a multi-GNSS orbit geometry. According to Feairheller and Clark (2006), the GLONASS constellation disposes of 21 active plus three spare satellites on three orbital planes inclined by 64.8° w.r.t. the equator. In contrast, Hofmann-Wellenhof et al. (2008) describe the GPS orbit constellation to consist of 21 satellites plus three spares on six orbital planes inclined by 55° w.r.t. the equator, and the Galileo orbit constellation to contain 27 operational plus three spare satellites on three planes inclined by 56° w.r.t. the equator. Therefore, five signal directions resp. five visible satellites correspond to a pessimistic GPS setting e.g. with site specific shadowing. Eight signal directions per site may be considered as a typical GPS setting at a GNSS permanent site, at which about 30 % of the total number of GPS satellites is visible at a time. Similarly, a total number of 20 signal directions per site corresponds to a visibility of about 30 % of the total number of satellites at a time within a multi-GNSS constellation composed of GPS, GLONASS, and Galileo. For each of the mentioned numbers of sites and numbers of visible satellites, 48 signal direction samples are defined. Given the GPS repeat cycle of about one day, the number of 48 signal direction samples is chosen in order to emulate about half-hourly orbit samples. Using the orbit characteristics mentioned above, such synthetic half-hourly satellite positions are approximated by means of circular orbits.

## 3.4 Study area and data sets

For each of the described observing geometry settings, synthetic SWD observations as input for the tomographic system are deduced from one single WRF simulation covering an about $200 \, \text{km} \times 200 \, \text{km}$ large area centered around the longitude and latitude $(\lambda, \varphi) = (8.15°, 49.15°)$ at a $900 \, \text{m}$ horizontal resolution. The vertical resolution increases with height, ranging from about $50 \, \text{m}$ to about $500 \, \text{m}$. As schematically illustrated in Figure 1, for each synthetic GNSS site, this is done by means of averaging the refractivity information of all WRF cells situated within the defined tomographic voxels, a direct raytracing within these tomographic voxels, and adding together the SWD along each signal direction within the tomographic voxels

using Equation 3. As the voxels are limited by ellipsoidal upper and lower borders, the raytracing is performed according to
Perler (2011).
The horizontal distribution of the synthetic GNSS sites within the URG study area is shown in Figure 2. The signal direc-
tions result from selecting at random the defined number of signal directions from a synthetic multi-GNSS orbit constellation
composed of GPS, GLONASS, and Galileo. Both signals entering the study area on its top and on its side are included.
From WRF, simulations of water vapor mixing ratio, temperature, pressure, and geopotential height are available at a $900\,\mathrm{m}$
horizontal resolution for generating the synthetic GNSS SWDs within the $95 \times 99\,\mathrm{km}^2$ large study area situated in the URG
as shown in Figure 2. The topography within the Rhine Valley is flat. Height differences mainly occur at the foot of the Black
Forest mountain range. The height difference between the highest and the lowest synthetic GNSS site used for this study is
about $494\,\mathrm{m}$.

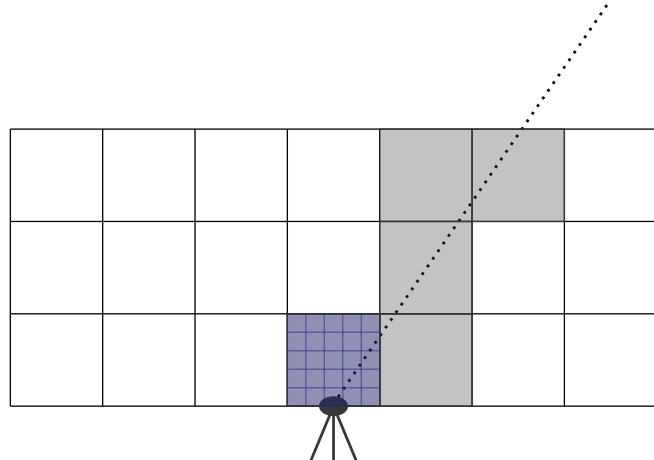

**Figure 1.** Schematic illustration of the generation of synthetic GNSS SWDs according to Heublein (2019): Within each tomographic voxel (grey), the WRF refractivities of all those WRF cells (blue) situated within that voxel are averaged. A direct raytracing along the considered signal direction then yields the SWD introduced into the tomographic system.

## 4  Results
For the most humid acquisition date (27 June 2005) for which WRF simulations were provided for this research and for an
exemplary voxel in the lower middle of the lowest voxel layer, Figure 3 shows that variations in the SWD signal directions
available within the tomographic system cause variations in the estimated refractivities. The magnitude of the absolute wet
refractivity values during the analyzed weather conditions ranges from $0\,\mathrm{ppm}$ to $74\,\mathrm{ppm}$.. As variations in the signal direc-
tions imply a change in the observed atmospheric volume, these variations in the estimated refractivities seem obvious. Yet,
Figure 3 illustrates that the variations in the refractivity estimates vary with the selected solution strategy. When considering
many sites and many signal directions per site (e.g. at least 27 sites and at least 15 signal directions), the difference between

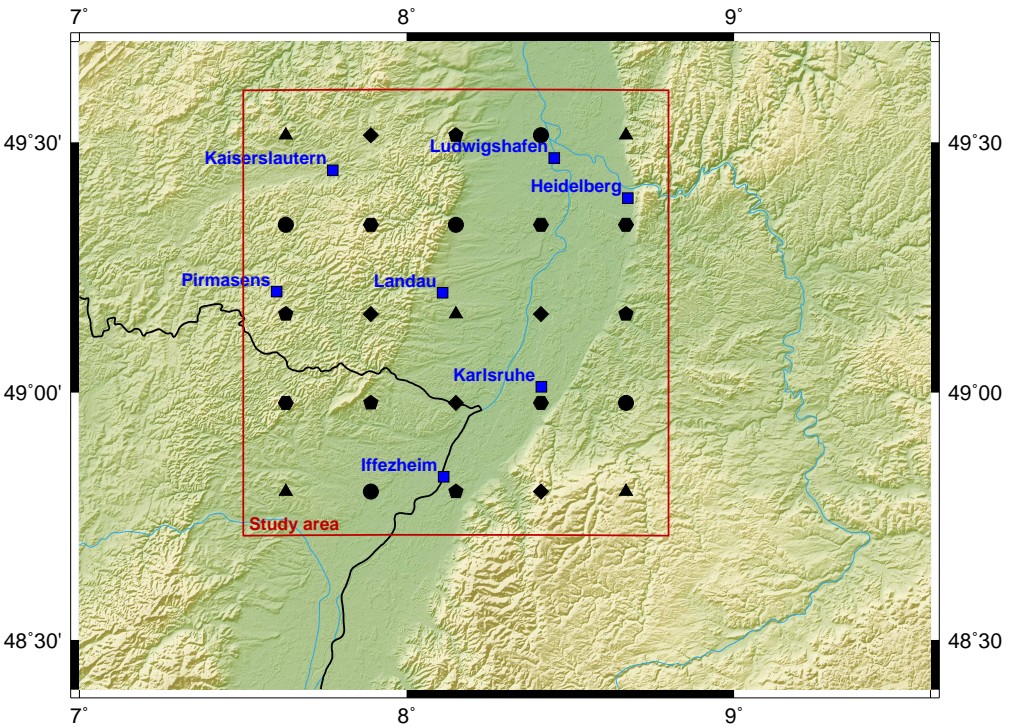

**Figure 2.** Distribution of the seven GNSS permanent sites (blue squares) as well as of the five to 25 additional, synthetic sites (black symbols) within the URG study area. The additional, synthetic sites are distributed within a grid that uniformly covers the study area. Triangles, pentagons, hexagons, diamonds, and circles represent the first, second, third, fourth, and fifth group of five additional sites each.

the CS refractivity estimates and the WRF refractivity of the considered voxel approaches zero for most samples. However,
e.g. for 27 sites and 20 signal directions per site, there are some samples in which the CS based refractivity estimate differs
from the WRF refractivity by up to $3.3\,\mathrm{ppm}$. I.e. for many signal directions, CS is able to accurately and precisely reconstruct
the voxel's refractivity, but for some signal directions, the voxel's refractivity estimate does not match well with the voxel's
validation refractivity from WRF. In contrast, in the case of few sites and few signal directions per site (e.g. twelve sites and
ten signal directions per site), LSQ yields refractivity estimates differing from $-5.9\,\mathrm{ppm}$ to $-0.7\,\mathrm{ppm}$ from the WRF refractiv-
ity, while the CS refractivity estimates differ much more from the WRF refractivity (differences of $-42.9\,\mathrm{ppm}$ to $26.9\,\mathrm{ppm}$).
Consequently, when investigating the observing geometry's effect on the quality of the tomographic reconstruction, the chosen
solution strategy as well as the effect of varying signal directions absolutely need to be taken into account. Therefore, in this
research, a representative set of 48 half-hourly samples of synthetic GNSS orbits is considered in order to analyze the observing
geometry's effect on the tomographic reconstruction quality.

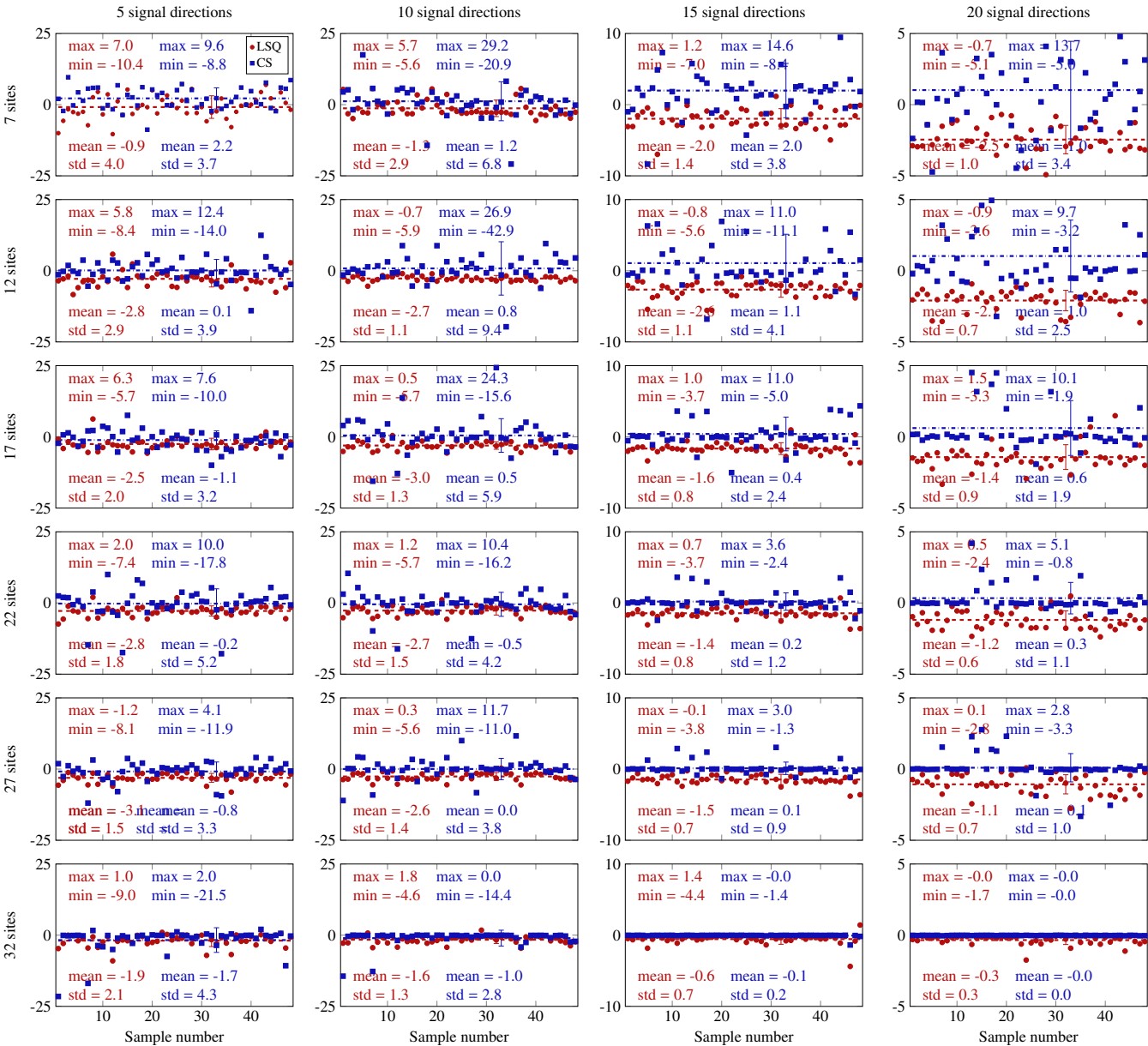

**Figure 3.** Absolute differences between estimated refractivity and the WRF refractivity in ppm for an exemplary voxel in the lower middle of the lowest voxel layer for the 48 samples of each investigated observing geometry setting. The two left columns dispose of an ordinate ranging from $-25\,\text{ppm}$ to $25\,\text{ppm}$, the third column plots the differences within the range $-10\,\text{ppm}$ to $10\,\text{ppm}$, and the right column plots the differences within the range $-5\,\text{ppm}$ to $5\,\text{ppm}$. The legend in the upper left subplot holds for all subplots: red circles stand for LSQ results, while blue squares represent the CS results. In each subplot, the minimum and maximum absolute differences in ppm of the LSQ resp. CS refractivity estimate w.r.t. the WRF input refractivities and the mean and the standard deviation over all samples are given in red resp. blue. Moreover, the mean and the standard deviation over all samples are indicated, for LSQ by a red dashed line resp. for CS by a blue dashdotted line and by errorbars in the corresponding colors.

As expected, an increased number of sites and an increased number of signal directions per site, in general, decrease the
mean of the absolute difference (called mean difference in the following) and the standard deviation of the difference between
estimated refractivities and WRF refractivities. Yet, as shown in Figure 4 averaged over all voxels, in the case of a LSQ solution
to the tomographic system, the mean difference decrease by means of introducing more SWD estimates into the tomographic
reconstruction is much smaller than that in the case of a CS solution. When averaged over 48 samples per observing geometry,
introducing more SWD estimates improves the mean difference by up to $1.3\,\mathrm{ppm}$ resp. $1.9\,\mathrm{ppm}$ (maximum improvement
observed for 20 resp. 15 signal directions per site in the case of LSQ resp. CS).

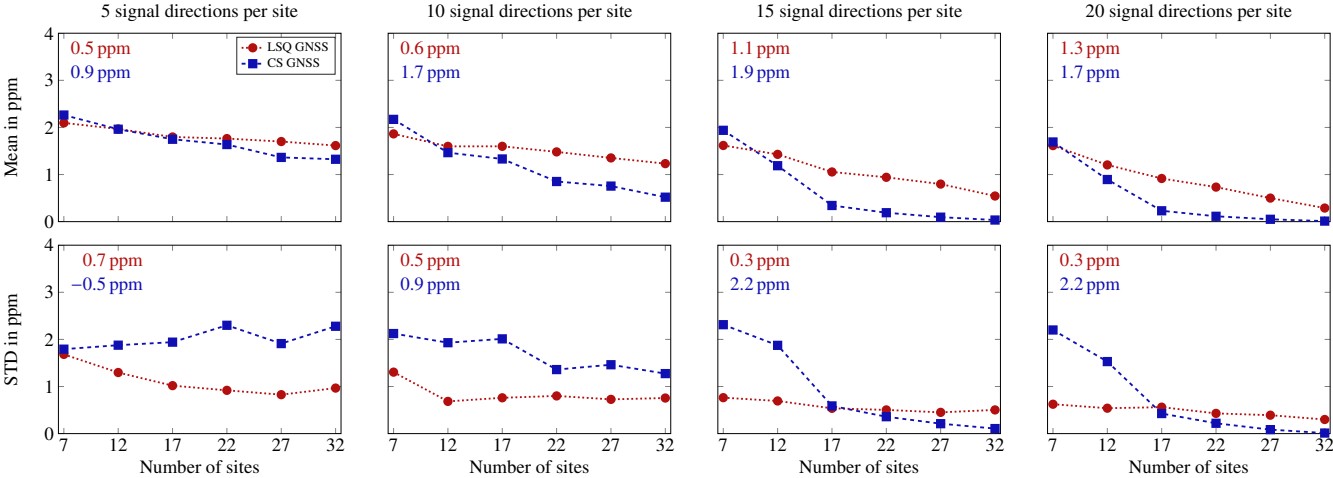

**Figure 4.** Averaged over all voxels, this figure shows the mean of the absolute difference and the std of the difference between estimated
refractivities and WRF refractivities in $\mathrm{ppm}$, deduced from 48 samples of each investigated observing geometry setting composed of a certain
number of synthetic GNSS sites and various numbers of signal directions per site. The dashed resp. dotted lines serve for better following
the variation of the represented quantities with the number of sites, but only the discrete values indicated by the markers should be evaluated.
The legend in the upper left subplot holds for all subplots. In each subplot, the improvement by introducing 32 sites instead of seven sites is
given in red resp. blue. Degradations are given with a minus sign.

When investigating the standard deviation of the differences between estimated refractivities and WRF refractivities for the CS
case, considering an increased number of synthetic GNSS sites only, while keeping a constant number of five signal directions
per site, is not advantageous. However, as of 15 different signal directions per site, a clear improvement in standard deviation
is visible when increasing the number of sites in the tomographic setting solved by means of CS. Independently of the number
of sites, for realistic GPS-like observing geometry settings with five to ten signal directions per site, the LSQ refractivity esti-
mates are more precise than the CS refractivity estimates. In contrast, as of 15 signal directions per site, the CS solution yields
more accurate and more precise refractivity estimates than the LSQ solution if at least 22 sites are available. I.e. this study
shows that LSQ is less sensitive to the number of signal directions than CS. Therefore, we recommend to use LSQ for water
vapor tomography disposing of GPS-only observations and CS for water vapor tomography disposing of multi-GNSS SWD
estimates.
In the case of the maximum number of sites and the maximum number of signal directions per site (32 sites and 20 signal di-
rections per site), when averaged over the 48 considered samples per observing geometry, the mean difference and the standard
deviation of the LSQ resp. CS reconstruction attain values of about $0.3\,\mathrm{ppm}$ resp. $0.0\,\mathrm{ppm}$. Therefore, the number of sites and
the number of signal directions per site are of particular interest when aiming at a very accurate and very precise tomographic
reconstruction using CS.
For the given $5 \times 5 \times 5$ voxel discretization with horizontal voxel sizes of $19\,\mathrm{km}$ resp. $20\,\mathrm{km}$, the rule of thumb of Champollion
et al. (2004) requests the mean inter site distance to correspond to no more than $19\,\mathrm{km}$ to $20\,\mathrm{km}$. The results show that, using
the investigated synthetic data set with and at least 15 signal directions per site, the CS solution is able to more accurately and
more precisely reconstruct the atmospheric water vapor distribution than LSQ in the case of 22 sites within the $95 \times 99\,\mathrm{km}^2$
large study area, i.e. at a site density of about one site per $20.7 \times 20.7\,\mathrm{km}^2$ which is a bit lower than that required by the rule of
thumb of Champollion et al. (2004). I.e. if 15 signal directions are available per site, the rule of thumb can be transferred from
the LSQ solution to water vapor tomography to CS solutions.
Consequently, the following three main results are summarized from this study:
1. The rule of thumb of Champollion et al. (2004) can be transferred from LSQ to CS.
2. Based on site distributions obeying the rule of thumb of Champollion et al. (2004), CS needs a variety of at least 15

18         signal directions per site in order to estimate the 3D refractivity field more accurately and precisely than LSQ.

3. While LSQ seems to be less sensitive on the number of signal directions than CS. Therefore, CS should only be used in

20         the case of multi-GNSS SWD estimates yielding a variety of at least 15 signal directions per site.

**5   Discussion and outlook**
Section 4 states that the rule of thumb of Champollion et al. (2004) does not only hold for a tomographic solution based on
LSQ, but that it also ensures a good tomographic reconstruction in the case of a CS solution. Although this finding is based
on many different observing geometry settings, it only refers to a single voxel discretization and to a single study area with a
single topography and a single site distribution within that study area. As a consequence, this research mainly investigates the
validity of the rule of thumb of Champollion et al. (2004) for CS for the given study area, weather condition, and voxel dis-
cretization. For generalization, further tests should be performed that repeat the described methodology for other study areas,
weather conditions, and voxel discretizations, and for site distributions varying not only in the number, but also in the position
of the sites.
Moreover, as the presented approach only relies on a synthetic data set deduced from WRF, the synthetic SWDs introduced
within the tomographic system in this research are too optimistic, when compared to real GNSS SWD estimates. Therefore, the
conclusions drawn in Section 4 cannot necessarily be transferred to tomographic applications involving real SWD estimates.

In order to get a better idea on the transferability of the results, the analysis should be repeated based on real data, or the effect of adding different types of noise to the synthetic SWD estimates should be investigated (e.g. measurement and sensor noise and uncertainties resulting from the observing geometry). In the presented approach, instead of mapping ZWDs to the slant signal directions as in the case of a real GNSS processing, the synthetic SWD data set is computed based on a direct raytracing within the same voxels in which the tomographic reconstruction is thereafter performed. Yet, Heublein (2019) shows that this involves neglecting both a voxel discretization error and a mapping error committed in the case of real data.

Furthermore, Section 4 shows that the standard deviation of the difference between LSQ refractivity estimates and WRF refractivities is $6\%$ to $65\%$ smaller than that computed based on the CS refractivity estimates, if at most ten different signal directions per site are available. In contrast, in the case of a high number of sites and a high number of signal directions per site, the LSQ reconstruction is not able to yield as accurate and as precise estimates of the water vapor distribution as CS. I.e. when solving the tomographic system by means of LSQ, increasing the number of SWD signal directions improves the tomographic reconstruction quality less than when using CS. This may be due to the geometric smoothing constraints forming the basis of the LSQ solution. In the case of a small number of observations, the smoothing constraints ensure a smooth solution free of outliers that does not necessarily correspond to the prevailing atmospheric conditions. In the case of a high variety of observations, the smoothing constraints become less important w.r.t. the data fidelity term within the LSQ solution to the tomographic system, but they still effect the tomographic solution. Even in the case of a very high number of observations, the tomographic system cannot be solved in a pure data-driven way. Instead, the tomographic solution always takes into account to the chosen model assumptions, i.e. the LSQ solution always applies a certain amount of smoothing.

In addition, a low resp. a high number of signal directions chosen from a synthetic multi-GNSS constellation for the recommendation of LSQ resp. CS for GPS-only resp. for multi-GNSS water vapor tomography applications should not be to set equal to considering a real GPS-only resp. a real multi-GNSS setting. Choosing a small number of signal directions from a multi-GNSS constellation yields a higher variability in the signal directions than choosing the same small number of signal directions from a GPS-only constellation. Since a high number of signal directions showed to be of particular importance in the case of a CS solution, the quality of the refractivity estimates deduced using CS may decrease, if real GPS-only signal directions are chosen.

Finally, future research should analyze in more detail which signal directions are necessary in a LSQ resp. CS based water vapor tomography in order to well reconstruct the refractivities of as much voxels as possible. A two-step CS LSQ may then first yield accurate refractivity estimates for most voxels by means of CS and then use the geometric smoothing constraints applied in LSQ to improve the refractivity estimates of those voxels in which CS yields inaccurate refractivity estimates even if a high number of sites and a high number of signal directions per site are available.

*Author contributions.* Marion Heublein developed the theory and performed the computations. Patrick Erik Bradley and Stefan Hinz verified the analytical methods and supervised the findings of Marion Heubleins work.

*Competing interests.* Within the last years, Marion Heublein and Stefan Hinz collaborated with the Signal Processing in Earth Observation (SiPEO) team of the German Aerospace Center and with Giovanni Nico (Consiglio Nazionale delle Ricerche Bari, Italy) and Pedro Benevides (University of Lisbon, Portugal).

*Acknowledgements.* The first author was supported by a scholarship of the Deutsche Telekom Stiftung. Thanks to Franz Ulmer (formerly at the German Aerospace Center at the Remote Sensing Technology Institute) for providing WRF data. The authors acknowledge support by the KIT-Publication Fund of the Karlsruhe Institute of Technology.

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
