# Peer review of "Observing geometry effects on a GNSS based water vapor tomography solved by Least Squares and by Compressive Sensing"

_Annales Geophysicae, 2019_

## Referee Comment (RC1) · Anonymous Referee #1 · 22 Aug 2019

In general i find the manuscript well written. I have some comments (see below). Some of them can be regarded major.

comments:

Abstract, L2: '... neutrospheric water vapor tomography...' please remove here and everywhere in the manuscript the word 'neutrospheric'.

Abstract, L5: '... The novelties of this research are 1) the comparison of the observing geometry's effects on the tomographic reconstruction accuracy when using LSQ resp. CS for the solution of the tomographic system and 2) the investigation of the effect of the signal directions' variability on the tomographic reconstruction.' I do not think that

these are novelties. Please see e.g. Bender et al. 2011, Zhao et al. 2019 etc. They also study impact of observation geometry. Also, see comment below.

Page1, L21: '... Therefore, a precise knowledge of the water vapor field is required for accurate positioning and deformation monitoring using GNSS and InSAR...' I do not agree. With the GNSS you estimate both position and state of the troposphere, i.e. you estimate SWD.

Page1, L22: '... However, the atmospheric water vapor distribution is difficult to model because it is highly variable in time and space.' This is true. Therefore, how can it be that you are trying to reconstruct it with 5 layers in the vertical only? This does not make sense to me. Explain to me the purpose of such water vapor field. It can not be weather forecasting. For example, i am sure that the number of vertical levels of the WRF (the weather research and forecasting) model you make use of is » 5.

Page 2, L11:'.. Yet, even after regularization, the observing geometry composed e.g. of the number and the geographic distribution of the GNSS sites, the SWD signal directions, and the voxel discretization still effects the quality of the tomographic solution.' With regards to the observation geometry you should mention the following papers:

Bender et al. 2011, GNSS water vapour tomography - Expected improvements by combining GPS, GLONASS and Galileo observations, Advance in Space Research

Zhao et al. 2019, Accuracy and reliability of tropospheric wet refractivity tomography with GPS, BDS, and GLONASS observations, Advance in Space Research

The conclusion from the first paper is '...'The reconstruction quality could not be improved considerably using the currently available technique...' Likewise, the conclusion from the second paper is '...Tomographic results also show that multi-GNSS observations can increase the accuracy of 3-d wet refractivity reconstruction but not as well as was expected when using currently available techniques...'

It appears that the problem in gnss tomography is not the observation geometry (!?).

Please start a discussion here.

Page 4, Line 8: equation 1, define dl.

Page 5, Line23: "...As Hscale is essential for defining an exponential decay with height, its value is determined within the solution of the tomographic system from a set of realistic values for Hscale between 1000 m and 2000 m ." I do not understand this sentence. What is 'realistic'? Do you estimate this parameter in your lsq? Let me suggest something: if the wet refractivity follows equation 10 (a single scale height), then it is absolute sufficient to measure the ZWD (zenith wet delay) and the refractivity at the station N0 to retrieve the wet refractivity profile. The reason is that ZWD = N0* Hscale so that Hscale=ZWD/N0. There is no need for SWDs. Please start a discussion here.

Page 8, Line 7: "...From WRF, simulations of water vapor mixing ratio, temperature, pressure, and geopotential height are available at a 900m spatial resolution.." provide a reference for WRF. Do i understand it correctly: the horizontal resolution of WRF is 900m?

Figure 3: i can not read the numbers. please increase the font size.

Figure 4 : "...in order to make the Euler refractivity decay with height visibly similar to the Euler letters used in the Compressive Sensing solution for modeling the refractivity decay with height ..." Could you please add in the Figure a wet refractivity profile with a single scale height Hscale of say 2km for comparison? Thank you.

Figure 5: please increase the font size.

---

## Referee Comment (RC2) · Anonymous Referee #2 · 28 Aug 2019

Although the manuscript is overally well written and generally contains work which is worthy to be published, I have some comments and questions including three major ones. I would like to ask the authors to adress all of them.

Major comments:

Section 2: Related work

I miss references to some of the important works related to GNSS tomography. To be more specific: - I mainly miss the work of Bender and Raabe (2007) dealing with preconditions to GNSS tomography and Bender et al. (2011) presenting evaluation of benefit using multi-GNSS tomography instead of GPS-only tomography. Paper by

[Figure]

Bender et al. (2009) is also related to the topic of reviewed manuscript.

- P2L22: i.e. Rohm et al. (2014) also used a Kalman filter for GNSS tomography - a novel and interesting approach for a geometric discretization of the tomographic network can be found in Ding et al. (2018)

Section 3.3:

- P6L28: I do not understand why you selected only 5 vertical layers in your study. I can hardly imagine any practical usage of such a coarse information about wet refractivity (water vapor) vertical distribution - i.e. your boundary layer has thickness of 1300 m while according to Seidel (2002), under standard weather conditions 50% of all water vapor is distributed below the height of 1.5 km above the surface. You mention that the boundary layer thickness is set to 1300 m in order to that signals pass the neighbouring voxel within the same height layer. Since I haven't seen this requirement in any other GNSS tomography study, could you please give reasons for it? In this regard I want to also ask why you have selected cut-off elevation angle of $7°$ while nowadays good quality slant delays could be obtained down to $5°$ or even $3°$?

According to results of other existing GNSS tomography studies (which many of them you cite in your manuscript), using between 10 and 15 vertical layers is definetely feasible. Therefore I would like to ask you to repeat your study with a higher number of vertical layers (at least 10) and provide the new results in the manuscript since I guess this setting can influence the results.

Section 4:

I have some questions and comments related to results presentation:

- P8L13: you write that figure 3 shows results "for an exemplary voxel of the lowest voxel layer". Which voxel is the selected exemplary one? The center one in 5x5 network? How have you selected right this voxel? Have you realized an overal evaluation across all the voxels? If yes, what were its results? Did you find (significant) differences among

individiual voxels or not? If not, I encourage you to do it and provide the results in the manuscript. Have you also studied the distribution of differences at various height levels? If yes, what were the results? I.e.: did LSQ or CS provided better/worse results either in boundary layer or at top of the troposphere? What was the impact of using more signal directions/GNSS sites? If you have not done such an evaluation, I again strongly encourage you to do it and present the results.

- P9L10: I know that you state in your conclusions that your findings are valid only for your limited study and encourage (yourself) to do more work, however I am afraid that one 48 h session can hardly provide enough data for a reasonable and trustworthy evaluation. Firstly, you should provide information on what period it was (summer, winter season), what were meteorological conditions during it and why you have selected it. Secondly, I would recommend to use at least two different periods since the results can be related to weather conditions and I encourage you to do so for your next version of the manuscript.

Minor comments:

- P2L11: you mention various applications of Compresive Sensing (CS) in this paragraph and some of its characteristics in the following paragraph, however the reader can be interested in a basic description of CS principles. Although you probably have a detailed description of CS in your cited papers (Heublein et al., 2018 and Heublein, 2019), I think it would be worthy to provide a short one (excluding the formulas which you have in Section 3.2) also here.

- P5L11: you mention a usage of "prior knowledge from surface meteorology". Could you be more specific on this? What exactly do you use for this purpose and what is the source of meteorological data (blind model, numerical weather prediction model, in-situ observations)?

- P5L20: just an information which you probably already know: although the water vapour (wet refractivity) standardly decreases exponentially with increasing height

above the surface, inversions in its vertical profile at various heights can commonly occur. And GNSS tomography solution should be ideally able to reconstruct such inversions.

- P7L0, Table 1: I wonder if it is worthy to keep the table in the manuscript since most of the information is also given in the paragraph below the table. I would like to warn you that GPS is using a basic constellation of 24+3 with a maximum possible number of 36 satellites since 2011, see i.e. https://www.gps.gov/systems/gps/space/. The 21+3 constellation stated from Hofmann-Wellenhof et al., 2008 was a previous one.

- P7L11: is there any reason why do you omit the chinese global GNSS system BeiDou in your study?

- P7L21: can you describe how do you define the "48 signal direction samples"? Using a real observation characteristic of satellites from individual GNSS systems or using something else, i.e. a regular azimuth/elevation spacing for signals?

- P8L7: could you provide at least basic information about a) raytracing technique which was used to compute slant delays? At least a reference should be given; b) WRF model used for your study - i.e. information about who operates this WRF model, for which area, what horizontal/vertical resolution is used, what is the source of initial and boundary conditions, etc.

- P9L4: probably a verb is missing in the sentence in the first half of the line

- P10 Figure 3: the scale of y axis is -25 ppm to +25 ppm, not -20 ppm to +20 ppm as you write in the figure caption. I suggest to provide mean and standard deviation values in numbers as the min/max value. They will be definetely better readable then. You should also increase font size of all text inside the figure.

- P11 Figure 4: I wonder if the figure 4 brings any useful information for the reader (you do not even describe its content in the manuscript). If you want to keep it, please provide information what you want to show with it (in my sense all the vertical profiles

are very similar to each other and are of an expected shape) and improve it (i.e. how does the voxel numbering works? Where is voxel 16 and where is voxel 17? For which date/period is the figure valid?).

- P11L1: I think the first sentence of the paragraph is completely expectable and logical and could be therefore deleted.

- P11L17: What exactly do you want to say with your sentence "I.e. this study recommends the use of LSQ resp. CS for water vapor tomography disposing of GPS-only resp. of multi-GNSS SWD"? That LSQ is recommended for GPS-only and CS for multi-GNSS tomography? If yes, please reformulate the sentence since it is not fully clear and can be confusing for the reader.

- P12 Figure 5: increase the font size of all the text in the figure (especially the text with "improvements" information)

- P12L15: could you please namely repeat here which of your tested number of stations corresponds to the rule of thumb of Champollion et al. (2004)?

- P13L1: I would rather write that LSQ seems to be not as sensitive on number of signal directions as CS than provide any "recommendation"

- P13L8: I am sure further tests should also be realized for various weather conditions and for various vertical distributions of vertical layers (changing total number of vertical layers and/or their heights) - please see my major comments in this regard.

- P13L29: Could you please briefly inform the reader why "Even in the case of a very high number of observations, the tomographic system cannot be solved in a pure data-driven way."?

- P14L6: I struggle to understand what you want to say with your last paragraph. Could you please reformulate it? At least I do not understand what does the "in order to well the refractivities of as much voxels as possible" means.

Whoops! Something went wrong.

References

- Bender, M., Raabe, A. Preconditions to ground based GPS water vapour tomography, Annales Geophysicae, Vol. 25, pp. 1727-1734, 2007

- Bender, M., Dick, Wickert, J., Ramatschi, M., G., Ge, M., Gendt, G., Rothacher, M., Raabe, A., Tetzlaff, G. Estimates of the information provided by GPS slant data observed in Germany regarding tomographic applications, Journal of Geohysical Research, Vol. 114, D06303, doi:10.1029/2008JD011008, 2009

- Bender, M., Stosius, R., Zus, F., Dick, G., Wickert, J., Raabe, A. GNSS water vapour tomography – Expected improvements by combining GPS, GLONASS and Galileo observations, Advances in Space Research, Vol. 47, Issue 5, pp. 886-897, 2011

- Ding, N., Zhang, S., Wu, S., Wang, X., Kealy, A., and Zhang, K.: A new approach for GNSS tomography from a few GNSS stations, Atmos. Meas. Tech., 11, 3511–3522, doi:10.5194/amt-11-3511-2018, 2018

- Seidel, D. J. Water Vapor: Distribution and Trends. The Earth System: Physical and Chemical Dimensions of Global Environmental Change, John Wiley & Sons, Ltd, 2002

- Rohm, W., Zhang, K., Bosy, J. Limited constraint, robust Kalman filtering for GNSS troposphere tomography, Atmos. Meas. Tech., 7, 1475–1486, doi:10.5194/amt-7-1475-2014, 2014

---

## Author Comment (AC1) · 26 Sep 2019

Dear Referee #1,

We would like to thank you for your valuable comments and deeply appreciate the time and effort spent on the review. The provided comments are very helpful and we thoroughly revised our manuscript. We hope that the revised version of the manuscript is acceptable for publication in the Annales Geophysicae and look forward to hearing from you soon.

[Figure]

Attached, you can find the revised manuscript and our answers to your comments. The respective changes in the manuscript are color-coded: green for referee #1 and blue for referee #2.

Yours sincerely,
Marion Heublein

On behalf of the co-authors Patrick Erik Bradley and Stefan Hinz

Please also note the supplement to this comment:
https://www.ann-geophys-discuss.net/angeo-2019-87/angeo-2019-87-AC1-supplement.zip

---

## Author Response (AR1)

**angeo-2019-87-RC1: Answers to the referees' comments**

Dear Editors and Reviewers,

We would like to thank you for your valuable comments and deeply appreciate the time and effort spent on the reviews. The provided comments are very helpful and we thoroughly revised our manuscript. We hope that the revised version of the manuscript is acceptable for publication in the Annales Geophysicae and look forward to hearing from you soon.

Yours sincerely,
Marion Heublein

On behalf of the co-authors Patrick Erik Bradley and Stefan Hinz

**General remark**

The comments of the two reviewers are answered below. The respective changes in the manuscript are color-coded: green for referee #1 and blue for referee #2.

**Referee #1: Modifications & answers highlighted in green**

1. Abstract, L2: '... neutrospheric water vapor tomography...' please remove here and everywhere in the manuscript the word 'neutrospheric'.
   We removed the word 'neutrospheric' within the whole manuscript.

2. Abstract, L5: '... The novelties of this research are 1) the comparison of the observing geometry's effects on the tomographic reconstruction accuracy when using LSQ resp. CS for the solution of the tomographic system and 2) the investigation of the effect of the signal directions' variability on the tomographic reconstruction.' I do not think that these are novelties. Please see e.g. Bender et al. 2011, Zhao et al. 2019 etc. They also study impact of observation geometry. Also, see comment below.
   So far, the observing geometry's effects on the tomographic reconstruction accuracy have only been investigated for other reconstruction algorithms than for CS. Therefore, 1) comparing the observing geometry's effects on the accuracy of a LSQ reconstruction and on a CS reconstruction is a novelty. Indeed, there have been previous studies on the effect of the observing geometry on the tomographic reconstruction accuracy of e.g. LSQ based water vapor tomography approaches. Yet, the observing geometry can be considered to be composed of many different aspects. As the previous studies did not focus on the sensitivity of the reconstruction algorithm w.r.t. the signal directions' variability, it is a novelty

to 2) investigate the effect of the signal directions' variability on the tomographic recon-
struction. We added the two mentioned papers of Bender et al. 2011 and Zhao et al. 2019
to the state of the art.

3. Page1, L21: '... Therefore, a precise knowledge of the water vapor field is required for ac-
curate positioning and deformation monitoring using GNSS and InSAR...' I do not agree.
With the GNSS you estimate both position and state of the troposphere, i.e. you estimate
SWD.
Indeed, with GNSS, you can estimate both position and state of the troposphere. We re-
formulated the sentence to "Therefore, a precise knowledge of the water vapor field e.g. is
required for accurate deformation monitoring using InSAR." Based on a good knowledge
on the water vapor distribution, the different phase components in InSAR processing can
e.g. be more easily be separated.

4. Page1, L22: '... However, the atmospheric water vapor distribution is difficult to model
because it is highly variable in time and space.' This is true. Therefore, how can it be
that you are trying to reconstruct it with 5 layers in the vertical only? This does not make
sense to me. Explain to me the purpose of such water vapor field. It can not be weather
forecasting. For example, i am sure that the number of vertical levels of the WRF (the
weather research and forecasting) model you make use of is » 5.
Thanks for this remark. The purpose of our research is to compare a new methodology
for tomographic reconstruction, namely CS, with existing LSQ tomography approaches,
in order to better understand the characteristics of both methodologies. Therefore, in this
publication, we only focus on five layers and we only use synthetic data sets, although
WRF provides much finer height levels. The presented research of course needs to be
extended to more height layers and to real data in the future.

5. Page 2, L11:'.. Yet, even after regularization, the observing geometry composed e.g. of
the number and the geographic distribution of the GNSS sites, the SWD signal directions,
and the voxel discretization still effects the quality of the tomographic solution.' With
regards to the observation geometry you should mention the following papers: Bender
et al. 2011, GNSS water vapour tomography - Expected improvements by combining
GPS, GLONASS and Galileo observations, Advance in Space Research Zhao et al. 2019,
Accuracy and reliability of tropospheric wet refractivity tomography with GPS, BDS, and
GLONASS observations, Advance in Space Research. The conclusion from the first paper
is '...'The reconstruction quality could not be improved considerably using the currently
available technique...' Likewise, the conclusion from the second paper is '...Tomographic
results also show that multi-GNSS observations can increase the accuracy of 3-d wet re-
fractivity reconstruction but not as well as was expected when using currently available
techniques...' It appears that the problem in gnss tomography is not the observation ge-
ometry (!?). Please start a discussion here.
We mentioned the two papers in the related work section.
Indeed, Bender et al. 2011 states that the "information provided by a network of GPS
or Galileo receivers is highly variable in space and time" and that "the situation can be
considerably improed by combining the observations of different GNSS" and finally realizes that "the reconstruction quality could not be improved substantially by combining the observations of different satellite systems". Moreover, Zhao et al. 2019 summarize that "although multi-GNSS can provide more observations, it cannot improve the quality of the tomography as was expected". However, Bender et al. 2011 also state that the results of a tomographic reconstruction do not only depend on the GNSS observations, "but also on the ill-posedness of the problem, the reconstruction algorithm, and several assumptions made for stabilise the inversion."

From these two publications, we would not generally conclude that the problem in GNSS tomography is not the observing geometry. Instead, we would conclude that using multi-GNSS observations in a water vapor tomography solved by the iterative reconstruction techniques (Bender et al. 2011) or by singular value decomposition (Zhao et al. 2019) does not improve the reconstruction accuracy, when compared to a GPS only solution. In our research, however, we neither focus on iterative reconstruction techniques nor on solving the tomographic system using singular value decomposition. Moreover, we do not focus on a possible accuracy improvement obtained using multi-GNSS observations instead of GPS only observations. Instead, we focus on comparing the two reconstruction algorithms LSQ and CS (including their assumptions made for stabilizing the inversion) w.r.t. the LSQ resp. CS sensitivity to variations in the ray directions.

6. Page 4, Line 8: equation 1, define dl.
   Thanks for the remark. We defined dl.

7. Page 5, Line23: "...As Hscale is essential for defining an exponential decay with height, its value is determined within the solution of the tomographic system from a set of realistic values for Hscale between 1000 m and 2000 m ." I do not understand this sentence. What is 'realistic'? Do you estimate this parameter in your lsq? Let me suggest something: if the wet refractivity follows equation 10 (a single scale height), then it is absolute sufficient to measure the ZWD (zenith wet delay) and the refractivity at the station N0 to retrieve the wet refractivity profile. The reason is that ZWD = N0* Hscale so that Hscale=ZWD/N0. There is no need for SWDs. Please start a discussion here.
   Thanks for these questions and the suggestion to deduce the scale height from measurements of ZWD and the refractivity at the station. Unfortunately, when considering the Upper Rhine Graben GNSS network that we use in the case of real data, we usually do not dispose of surface meteorological measurements at the available GNSS permanent sites. I.e. we do not dispose of measurements of e.g. temperature, dew point temperature, and pressure at the GNSS sites, from which values of the refractivity at the station could be deduced. In the presented research, we could have computed the scale height based on the water vapor simulations from WRF. Yet, in order to keep the presented solution strategy for synthetic data transferable to real data, we only used WRF (at different seasons, times of day, or locations) in order to define the realistic values of Hscale to be situated between 1 km and 2 km. Based on this constraint, we then estimated Hscale from LSQ.

8. Page 8, Line 7: "...From WRF, simulations of water vapor mixing ratio, temperature, pressure, and geopotential height are available at a 900m spatial resolution.." provide a reference for WRF. Do i understand it correctly: the horizontal resolution of WRF is

900m?

Yes, you understood it correctly. The horizontal resolution of the available WRF simulations is 900 m. We added a reference for WRF.

9. Figure 3: i can not read the numbers. please increase the font size.
We increased the font size inside the figure.

10. Figure 4 : "...in order to make the Euler refractivity decay with height visibly similar to the Euler letters used in the Compressive Sensing solution for modeling the refractivity decay with height ..." Could you please add in the Figure a wet refractivity profile with a single scale height Hscale of say 2km for comparison? Thank you.
As the second referee wondered if this figure brings any useful information for the reader at all and as our goal was just to show the magnitude of the absolute wet refractivity values encountered during the analyzed weather conditions, we deleted the figure and now give the magnitude of the absolute wet refractivity values in the text.

11. Figure 5: please increase the font size
We increased the font size.

**Referee #2: Modifications & answers highlighted in blue**

Major comments:

1. Section 2: Related work I miss references to some of the important works related to GNSS tomography. To be more specific: I mainly miss the work of

   - Bender and Raabe (2007) dealing with preconditions to GNSS tomography and Bender et al. (2011) presenting evaluation of benefit using multi-GNSS tomography instead of GPS-only tomography.

   - Bender et al. (2009) is also related to the topic of reviewed manuscript.

   - P2L22: i.e. Rohm et al. (2014) also used a Kalman filter for GNSS tomography

   - a novel and interesting approach for a geometric discretization of the tomographic network can be found in Ding et al. (2018)

   Thanks for these remarks. We added the missing references.

2. Section 3.3:

   - P6L28: I do not understand why you selected only 5 vertical layers in your study. I can hardly imagine any practical usage of such a coarse information about wet refractivity (water vapor) vertical distribution - i.e. your boundary layer has thickness of 1300 m while according to Seidel (2002), under standard weather conditions 50 % of all water vapor is distributed below the height of 1.5 km above the surface. According to results of other existing GNSS tomography studies (which many of them you cite in your manuscript), using between 10 and 15 vertical layers is definetely feasible. Therefore I would like to ask you to repeat your study with a higher

number of vertical layers (at least 10) and provide the new results in the manuscript since I guess this setting can influence the results.

Thanks for these remarks. The purpose of our research is to compare a new methodology for tomographic reconstruction, namely CS, with existing LSQ tomography approaches, in order to better understand the characteristics of both methodologies. Therefore, in this publication, we only focus on five layers and we only use synthetic data sets, although WRF provides much finer height levels. The presented research of course needs to be extended to more height layers and to real data in the future.

- You mention that the boundary layer thickness is set to 1300 m in order to that signals pass the neighbouring voxel within the same height layer. Since I haven't seen this requirement in any other GNSS tomography study, could you please give reasons for it?

Champollion et al. (2005) states that the height layer thickness should be defined such that rays starting from one site are able to cross neighboring voxels. In their study, taking the site "at the center of the cell and a minimum elevation angle of $10°$, this implies a minimum thickness of the first layer of about 300 m." (They deal with about 17 sites in an only $20\,\mathrm{km} \times 20\,\mathrm{km}$ large study area; i.e. with a dense GNSS network.) They do not clearly state if the ray starting from one site at the center of a voxel should be able to cross the neighboring voxels of the same height layer or any neighboring voxel columns. Having some rays cross the neighboring voxel columns is essential for any water vapor tomography, because otherwise, vertical voxel layers could be simply interchanged and the vertical water vapor distribution could not be determined. For us, an additional motivation for enabling at least very low rays to cross the neighboring voxels of the same height layer is that this helps to avoid too similar rows in the design matrix. In order to well resolve the water vapor field, we want to produce many different rows in the design matrix.

- In this regard I want to also ask why you have selected cut-off elevation angle of $7°$ while nowadays good quality slant delays could be obtained down to $5°$ or even $3°$?

Thanks for this remark. You are right, that very low elevations are of benefit for any tomographic reconstruction. Therefore, using even lower elevations than $7°$ is attractive. In our research, the minimum elevation angle of $7°$ was set according to a comparable real data case within the same study area of a size of about $100\,\mathrm{km} \times 100\,\mathrm{km} \times 10\,\mathrm{km}$, in which only those rays were considered, that entered the study area on its top. As the horizontal extension of the study area is only $100\,\mathrm{km} \times 100\,\mathrm{km}$, rays with lower elevation angles than $7°$ would have hardly had the possibility to enter the study area at its top. However, you are right that nowadays, lower elevation angles and side rays should be included for any tomography.

3. Section 4: I have some questions and comments related to results presentation:

   - P8L13: you write that figure 3 shows results "for an exemplary voxel of the lowest voxel layer". Which voxel is the selected exemplary one? The center one in 5x5 network? How have you selected right this voxel? Have you realized an overal evaluation across all the voxels? If yes, what were its results? Did you find (significant)

differences among individiual voxels or not? If not, I encourage you to do it and provide the results in the manuscript. Have you also studied the distribution of differences at various height levels? If yes, what were the results? I.e.: did LSQ or CS provided better/worse results either in boundary layer or at top of the troposphere? What was the impact of using more signal directions/GNSS sites? If you have not done such an evaluation, I again strongly encourage you to do it and present the results.

The selected exemplary voxel is voxel 18, i.e. the fourth of five voxels of the middle voxel row. It was selected at random from all those ground voxels, for which no prior knowledge from surface meteorology was included.

An overall evaluation across all the voxels is shown in Figure 5 of the initial submission, as specified in the text and in the figure caption in the revised version of the manuscript. We also analyzed differences among individual voxels but did not find any significant differences among the individual voxels. We also studied the distribution of the differences at various height levels. Yet, LSQ or CS did not systematically provide better / worse results either in the boundary layer or at the top of the troposphere.

- P9L10: I know that you state in your conclusions that your findings are valid only for your limited study and encourage (yourself) to do more work, however I am afraid that one 48 h session can hardly provide enough data for a reasonable and trustworthy evaluation. Firstly, you should provide information on what period it was (summer, winter season), what were meteorological conditions during it and why you have selected it. Secondly, I would recommend to use at least two different periods since the results can be related to weather conditions and I encourage you to do so for your next version of the manuscript.

  Thanks for this comment. The results shown in the manuscript originate from WRF simulations for 27 June 2005 around 9:50 UTC, which is the most humid date (out of nine dates separated by 35 days each) for which we dispose of WRF simulations. W.r.t. the referee's comment that one 48 h session can hardly provide enough data for a reasonable and trustworthy evaluation, we would like to emphasize that not even a full 48 h session was analyzed. The synthetic ray directions correspond to synthetic satellite positions within 48 h, but we just use a single WRF simulation of 9:50 UTC of 27 June 2005 for all signal direction samples that we generated. This approach is motivated by the fact that we purely want to investigate the effect of the ray directions' variability. We do not want to have any side effects that might occur if we considered a real 48 h session with e.g. colder and dryer weather during night time (which might be more easy to reconstruct) and warmer and more humid weather during day time (that might be more difficult to reconstruct).

  In order to provide enough data for a reasonable evaluation, to our mind, it would not be enough to show results for more than one acquisition time. Instead, we should also investigate more than one study area (i.e. more than one location, topography, site distribution), because CS showed to be more sensitive to the observing geometry than LSQ.

Minor comments:

1. P2L11: you mention various applications of Compresive Sensing (CS) in this paragraph and some of its characteristics in the following paragraph, however the reader can be interested in a basic description of CS principles. Although you probably have a detailed description of CS in your cited papers (Heublein et al., 2018 and Heublein, 2019), I think it would be worthy to provide a short one (excluding the formulas which you have in Section 3.2) also here. We added some more explanations on CS in the methodology section, within the theory on the solution of the tomographic system. Moreover, we added a very short summary of the CS basic idea within the introduction.

2. P5L11: you mention a usage of "prior knowledge from surface meteorology". Could you be more specific on this? What exactly do you use for this purpose and what is the source of meteorological data (blind model, numerical weather prediction model, in-situ observations)?
We specified that the prior knowledge from surface meteorology results from in-situ observations of pressure, temperature, and dew point temperature, from which we compute a wet refractivity value at the surface.

3. P5L20: just an information which you probably already know: although the water vapour (wet refractivity) standardly decreases exponentially with increasing height above the surface, inversions in its vertical profile at various heights can commonly occur. And GNSS tomography solution should be ideally able to reconstruct such inversions.
Thanks for the remark. As we are aware of the possibility of inversions in the wet refractivity profile, we introduced both Euler letters and Dirac letters in the CS case. The Euler letters model an exponential refractivity decay and the Dirac letters should model any deviations from such an exponential refractivity decay.

4. P7L0, Table 1: I wonder if it is worthy to keep the table in the manuscript since most of the information is also given in the paragraph below the table. I would like to warn you that GPS is using a basic constellation of 24+3 with a maximum possible number of 36 satellites since 2011, see i.e. https://www.gps.gov/systems/gps/space/. The 21+3 constellation stated from Hofmann-Wellenhof et al., 2008 was a previous one.
Thanks for the remark, also w.r.t. the changed number of satellites since 2011. We deleted the table. In future research, we will simulate the satellite positions with the updated number of satellites.

5. P7L11: is there any reason why do you omit the chinese global GNSS system BeiDou in your study?
No, there is no special reason.

6. P7L21: can you describe how do you define the "48 signal direction samples"? Using a real observation characteristic of satellites from individual GNSS systems or using something else, i.e. a regular azimuth/elevation spacing for signals?
We define the 48 signal direction samples using half-hourly position samples of synthetic satellites circulating on circular orbits with the mentioned real orbit characteristics.

7. P8L7: could you provide at least basic information about

   a) raytracing technique which was used to compute slant delays? At least a reference should be given;
   We added a reference.

   b) WRF model used for your study - i.e. information about who operates this WRF model, for which area, what horizontal/vertical resolution is used, what is the source of initial and boundary conditions, etc.
   We specified the characteristics of the used WRF data and that the WRF model was run by Franz Ulmer during his work at DLR IMF.

8. P9L4: probably a verb is missing in the sentence in the first half of the line
   Thanks for the remark. We added the missing verb.

9. P10 Figure 3: the scale of y axis is -25 ppm to +25 ppm, not -20 ppm to +20 ppm as you write in the figure caption. I suggest to provide mean and standard deviation values in numbers as the min/max value. They will be definetely better readable then. You should also increase font size of all text inside the figure.
   Thanks for the attentive reading. We added the mean and the standard deviation values in numbers like the min and max values and increased the font size inside the figure.

10. P11 Figure 4: I wonder if the figure 4 brings any useful information for the reader (you do not even describe its content in the manuscript). If you want to keep it, please provide information what you want to show with it (in my sense all the vertical profiles are very similar to each other and are of an expected shape) and improve it (i.e. how does the voxel numbering works? Where is voxel 16 and where is voxel 17? For which date/period is the figure valid?).
    In P8L16 of the initial submission, we describe that we added the figure in order to better classify the magnitude of the difference values in Figure 3. Moreover, we added the figure in order to better illustrate the exponential decay of the wet refractivity with increasing height. Yet, you are right, the reader does not gain a lot of information by means of that figure. Therefore, we deleted it. In order to still be able to classify the magnitude of the differences, we now include absolute values of the wet refractivity in the text.

11. P11L1: I think the first sentence of the paragraph is completely expectable and logical and could be therefore deleted.
    We deleted the sentence.

12. P11L17: What exactly do you want to say with your sentence "I.e. this study recommends the use of LSQ resp. CS for water vapor tomography disposing of GPS-only resp. of multi-GNSS SWD"? That LSQ is recommended for GPS-only and CS for multiGNSS tomography? If yes, please reformulate the sentence since it is not fully clear and can be confusing for the reader.
    We reformulated the sentence.

13. P12 Figure 5: increase the font size of all the text in the figure (especially the text with "improvements" information)
We increased the font size.

14. P12L15: could you please namely repeat here which of your tested number of stations corresponds to the rule of thumb of Champollion et al. (2004)?
As stated on P12L6 of the initial submission, the rule of thumb requests a mean inter site distance of no more than 19 km resp. 20 km, if horizontal voxel sizes of 19 km resp. 20km are defined. Moreover, as stated on P12L9 of the initial submission, a total 22 sites yield a site density of about one site per $20.7\,km \times 20.7\,km$, which is a bit lower than that required by the rule of thumb of Champollion. For 27 sites, the rule of thumb is clearly fulfilled (18.7km mean inter site distance), for 22 sites, it is not yet fulfilled. We did not indicate a site number that exactly matches the 20km mean inter site distance required by the rule of thumb, because we did not test such a site number. In how far does this answer your question?

15. P13L1: I would rather write that LSQ seems to be not as sensitive on number of signal directions as CS than provide any "recommendation"
We reformulated the sentence.

16. P13L8: I am sure further tests should also be realized for various weather conditions and for various vertical distributions of vertical layers (changing total number of vertical layers and/or their heights) - please see my major comments in this regard.
We agree with this remark and reformulated the sentence.

17. P13L29: Could you please briefly inform the reader why "Even in the case of a very high number of observations, the tomographic system cannot be solved in a pure datadriven way."?
We slightly reformulated our explanation. The tomographic system cannot be solved in a pure data driven way because the solution of the tomographic system always involves some kind of regularization which is based on model assumptions and not only on the observed data (e.g. smoothing constraints.)

18. P14L6: I struggle to understand what you want to say with your last paragraph. Could you please reformulate it? At least I do not understand what does the "in order to well the refractivities of as much voxels as possible" means.
We reformulated the last paragraph.

References

1. Bender, M., Raabe, A. Preconditions to ground based GPS water vapour tomography, Annales Geophysicae, Vol. 25, pp. 1727-1734, 2007

2. Bender, M., Dick, Wickert, J., Ramatschi, M., G., Ge, M., Gendt, G., Rothacher, M., Raabe, A., Tetzlaff, G. Estimates of the information provided by GPS slant data observed in Germany regarding tomographic applications, Journal of Geohysical Research, Vol. 114, D06303, doi:10.1029/2008JD011008, 2009

3. Bender, M., Stosius, R., Zus, F., Dick, G., Wickert, J., Raabe, A. GNSS water vapour tomography – Expected improvements by combining GPS, GLONASS and Galileo observations, Advances in Space Research, Vol. 47, Issue 5, pp. 886-897, 2011

4. Ding, N., Zhang, S., Wu, S., Wang, X., Kealy, A., and Zhang, K.: A new approach for GNSS tomography from a few GNSS stations, Atmos. Meas. Tech., 11, 3511–3522, doi:10.5194/amt-11-3511-2018, 2018

5. Seidel, D. J. Water Vapor: Distribution and Trends. The Earth System: Physical and Chemical Dimensions of Global Environmental Change, John Wiley & Sons, Ltd, 2002

6. Rohm, W., Zhang, K., Bosy, J. Limited constraint, robust Kalman filtering for GNSS troposphere tomography, Atmos. Meas. Tech., 7, 1475–1486, doi:10.5194/amt-7-1475-2014, 2014

7. Champollion, C., Masson, F., Bouin, M. N., Walpersdorf, A., Doerflinger, E., Bock, O., and Van Baelen, J. (2005). GPS water vapour tomography: preliminary results from the ESCOMPTE field experiment. Atmospheric research, 74(1-4), 253-274.

---

## Author Response (AR2)

**angeo-2019-87-RC2: Answers to the referees' comments**

Dear Editors and Reviewers,

We would like to thank you again for your valuable comments and deeply appreciate the time and effort spent on the reviews. We hope that the current version of the manuscript is now acceptable for publication in the Annales Geophysicae and look forward to hearing from you soon.

In the following, we address all the provided comments and suggestions in detail.

Yours sincerely,
Marion Heublein

On behalf of the co-authors Patrick Erik Bradley and Stefan Hinz

**General remark**

The finalizing comments of the second reviewer are answered below. The respective changes in the manuscript are color-coded in blue.

**Referee #2: Modifications & answers highlighted in blue**

Major comment:

In case of major comments you unfortunatelly haven't repeated your processing using a vertical discretization with a higher number of layers as I recommended to you (i.e. at least with 10 vertical layers). I agree that your work is focused on evaluating the new methodology for GNSS tomography (CS), however the experiment still should be based on rational conditions under which the tomography can be providing some useful information about 3d water vapor field. I admit that in your revised version of the manuscript (namely in Section 5) you point out weaknesses of your study (including the discretization), however if you are really not able to repeat your processing using a more detailed vertical discretization (still based on sythetic slant delays and a single WRF simulation) I ask you to at least emphasize in the text the potential limitation of using only 5 vertical layers on the validity of the results. If this is done and my below given minor comments are solved, I can agree with the publication of your manuscript in Annales Geophysicae.
We emphasized the potential limitation of using only 5 vertical layers on the validity of the results.

Minor comments:

- P1L18: I am not sure if "three dimensional (3d) atmospheric water vapor distribution is essential for climate research". I would say that climatologists are rather interested in spatio-temporal regional and global trends in water vapor 2D distribution and can live without the 3d information. Please reformulate the sentence accordingly or provide a reference for your statement.
  We reformulated the sentence.

- P1L20: Can you provide a reference for your statement that "a precise knowledge of the water vapor field e.g. is required for accurate deformation monitoring using InSAR."? Water vapor does not influence only microwave signals, it causes a problem also i.e. in optical remote sensing
  We added a reference.

- P8L3: I think you can shortly note in the text that numbers of signal directions you use as GPS-only or multi-GNSS are a bit pesimistic and in reality one can expect higher numbers. Usually at permanent GNSS stations (from which most of them have excellent visibility over the sky) you have 8-12 observations for GPS-only, 12-18 for GPS+GLONASS and we can expect around 25-35 satellites once Galileo and BeiDou are fully operational. Regarding the BeiDou - you answered my question that "there is no special reason." why you have ommited this system from your work, however I think you should inform the reader that there exists also this 4th GNSS which can be also used to increase the number of signal directions in tomography applications.
  We informed the reader that BeiDou could also be used to increase the number of signal directions in tomography applications and that there may be up to twelve observations for GPS-only.

- P8: from L9 to L27 the text is repeated, I mean it is the same as text which you have from P7L24 to P8L9
  Thanks for your attentive reading. We deleted the repeated text.

- P8L28: in your section 3.4 you repeat some information which were already given and I think the rest of the information can (should) be provided within the section 3.3. Therefore I think sections 3.3 and 3.4 should be merged into one section to improve the readability of the manuscript.
  We merged the two sections into one in order to improve the readability of the manuscript and updated the title of the merged section.

- P10: what I see in figure 3 is: LSQ is everywhere except four boxes in the upper-left and lower-right corner providing better results than CS - although you summarize this in Section 5, I think you should mention it also here.
  We also added this information into the description of Figure 3.

- P10L9: I recommend to replace "absolutely need to" from your sentence "... the chosen solution strategy as well as the effect of varying signal directions absolutely need to be taken into account." with "should be taken into account". Your statement with "absolutely" is too strong in my opininon considering the limited dataset you use.
  Thanks for the remark. We replaced the terms as you proposed.

- P11, figure3: please check figure for 5 signal directions/27 sites. It has some problem with plotting the caption for CS mean/std.
  Thanks for the remark. We solved the plotting problem.

- P12: I am adressing my major comment for P8L13 from my previous review - in your reply you write: "We also analyzed differences among individual voxels but did not find any significant differences among the individual voxels. We also studied the distribution of the differences at various height levels. Yet, LSQ or CS did not systematically provide better / worse results either in the boundary layer or at the top of the troposphere." Please, add all this information to your manuscript to inform the reader about these results.
  We added all this information to our manuscript.

- P14L3: this is the first time in the manuscript you use term "ZWD", you should therefore provide its full meaning and ideally shortly desribe the relation between ZWD and SWD
  We defined the abbreviation ZWD and added a reference for mapping between ZWD and SWD.